# High-Altitude Stress Orchestrates mRNA Expression and Alternative Splicing of Ovarian Follicle Development Genes in Tibetan Sheep

**DOI:** 10.3390/ani12202812

**Published:** 2022-10-18

**Authors:** Wenhao Li, Weike Zeng, Xiayang Jin, Huiming Xu, Xingyan Fang, Zhijie Ma, Gangjian Cao, Ruizhe Li, Liuyin Ma

**Affiliations:** 1Academy of Animal Science and Veterinary Medicine, Qinghai University, Xining 810016, China; 2College of Forestry, School of Future Technology, Fujian Agriculture and Forestry University, Fuzhou 350002, China; 3College of Life Sciences, Fujian Agriculture and Forestry University, Fuzhou 350002, China

**Keywords:** ovary tissue, gene expression, transcription factor, alternative splicing, follicular development, high-altitude stress, Tibetan sheep

## Abstract

**Simple Summary:**

To achieve optimal growth performance and improved fertility in animals living on high plateaus, it is important to understand how high-altitude stress reduces fertility in females. This study analyzed the transcriptome dynamics of Tibetan sheep ovaries under high-altitude stress. High-altitude stress suppressed the expression of follicular development marker genes and impaired the luteinizing hormone/follicle-stimulating hormone signaling pathway. High-altitude stress also increased abnormally spliced isoforms of transcription factors and RNA processing factors. Therefore, high-altitude stress may reduce the fertility of Tibetan sheep by disrupting the normal expression/hormone signaling of follicular development genes. Further work is needed to decipher whether this phenomenon is a unique feature of Tibetan sheep or a general mechanism in animals under high-altitude stress.

**Abstract:**

High-altitude stress threatens the survival rate of Tibetan sheep and reduces their fertility. However, the molecular basis of this phenomenon remains elusive. Here, we used RNA-seq to elucidate the transcriptome dynamics of high-altitude stress in Tibetan sheep ovaries. In total, 104 genes were characterized as high-altitude stress-related differentially expressed genes (DEGs). In addition, 36 DEGs contributed to ovarian follicle development, and 28 of them were downregulated under high-altitude stress. In particular, high-altitude stress significantly suppressed the expression of two ovarian lymphatic system marker genes: *LYVE1* and *ADAMTS-1*. Network analysis revealed that luteinizing hormone (LH)/follicle-stimulating hormone (FSH) signaling-related genes, such as *EGR1*, *FKBP5*, *DUSP1*, and *FOS*, were central regulators in the DEG network, and these genes were also suppressed under high-altitude stress. As a post-transcriptional regulation mechanism, alternative splicing (AS) is ubiquitous in Tibetan sheep. High-altitude stress induced 917 differentially alternative splicing (DAS) events. High-altitude stress modulated DAS in an AS-type-specific manner: suppressing skipped exon events but increasing retained intron events. C_2_H_2_-type zinc finger transcription factors and RNA processing factors were mainly enriched in DAS. These findings revealed high-altitude stress repressed ovarian development by suppressing the gene expression of LH/FSH hormone signaling genes and inducing intron retention of C_2_H_2_-type zinc finger transcription factors.

## 1. Introduction

The growth and reproduction of animals are highly sensitive to changes in their surrounding environment [1]. Most animals use reproductive strategies such as seasonal reproduction to minimize the impact of adverse environmental factors on the reproduction [1]. However, high-altitude stress (hypobaric hypoxic stress, oxidative stress, or both) is an unavoidable environmental factor for domestic animals living in high altitude [1,2]. High-altitude stress affects the reproductive function of livestock, including sheep, resulting in significantly reduced fertility compared to their low-altitude counterparts [1].

More than 25 million people live above 2500 m above sea level (m.a.s.l), especially Qinghai–Tibetan and Andean high plateaus [1,2]. Sheep are a major economic resource for people living in high altitude [2]. Tibetan sheep are sheep that live on the Qinghai–Tibet Plateau (3000–5000 m.a.s.l, average height is ~4000 m.a.s.l). They offer wool, fur, meat, and milk for Tibetans and, thus, have important economic and agricultural value for the local community [3]. The number of Tibetan sheep in Qinghai Province accounts for one third of the total number of Tibetan sheep on the Qinghai–Tibet Plateau [4]. The annual output value of Tibetan sheep industry in Qinghai Province alone reaches CNY 12.70 billion or USD ~1.95 billion [4]. Although Tibetan sheep have experienced about 2000–2600 years to adapt to high-altitude environment [5], high-altitude stress still reduces their fertility. The Tibetan ewes only give birth to one lamb a year, and the mortality rate of neonatal lambs is ~6% [6]. Likewise, the lamb:ewe ratio has been reported to be only 0.4 in the Andean highlands [2]. Therefore, Tibetan sheep is a good breed for studying high-altitude stress-responsive mechanisms. However, the molecular basis of high-altitude stress in Tibetan sheep is largely unexplored, and the contribution of post-transcriptional regulation under high-altitude stress has not been characterized in sheep.

High-throughput RNA sequencing has been adopted to unveil the dynamics of gene expression in sheep ovarian tissues [7,8,9]. At the single-gene level, high-altitude stress induced changes in the expression of *IGF-I* and *IGF-II* transcript levels in the ovarian tissue of Andean plateau sheep [2]. However, the genome-wide transcriptome dynamics of ovarian tissue under high-altitude stress remain elusive. Furthermore, how altitude stress affects the regulation of gene expression at the post-transcriptional level is also largely unexplored. Alternative splicing (AS) is an important form of pre-mRNA processing in archaea, bacteria, and eukaryotes [10,11]. Alternative splicing facilitates diversification of the transcriptome and proteome by generating distinct mRNA isoforms [10]. Alternative splicing has been reported to involve cold stress, salt stress, and nutrient stress [12,13,14,15]. However, how AS contributes to high-altitude stress is largely unexplored.

In this study, we used high-throughput RNA-seq to understand the response of Tibetan sheep ovarian tissue to high-altitude stress. Transcriptional and post-transcriptional regulations of ovarian developmental genes were systematically analyzed. Differentially expressed gene analysis revealed that hormone-related transcription factors are pivotal regulators of the high-altitude stress regulatory network. Alternative splicing and differential alternative splicing analyses indicated that high-altitude stress induced AS events in an AS-type-specific manner. High-altitude stress induced retained intron but inhibited skipped exon events. Furthermore, high-altitude stress-related DAS events were mainly enriched in C_2_H_2_-type zinc figure transcription factors and RNA processing factors. Overall, the purpose of this study was to investigate the effects of high-altitude stress on the expression of ovarian development-related genes in Tibetan sheep at the transcriptional and post-transcriptional level.

## 2. Materials and Methods

### 2.1. Animal and Tissue Samples

Tibetan sheep ovary tissues under high-altitude stress were collected in July 2020 at the Zangbala Breeding Farm (average altitude 4400 m.a.s.l) in Yushu City (E 97°02′; N 33°00′), Qinghai Province, China. Likewise, low-altitude Tibetan sheep ovary tissues were collected in July 2020 from the Tibetan sheep breeding co-operative in Jintan Township (average altitude 3000 m.a.s.l), Haiyan County (E 100°98′; N 36°90′), Qinghai Province, China. According to the previous study [16], the ovaries of two-year-old nulliparous Tibetan sheep ewes have sexual maturity and complete ovarian function. Thus, all ovary samples were collected from two-year-old nulliparous ewes. For the high-altitude stress sample group, 6 ewes were randomly selected from 42 nulliparous Tibetan sheep ewes at two-year-old age. Similarly, a total of 6 ewes were also randomly selected from 90 two-year-old ewes as the low-altitude sample group. The left ovary of selected ewes was collected, and the blood and stains on the surface of the ovarian tissue were quickly washed twice with 1 × PBS prepared with 4 °C precooled RNase-Free water. The surface liquid was quickly sucked dry with a dust-free paper towel, and the tissues were quickly stored in a liquid nitrogen cylinder for transportation.

The experiment was conducted during the local slaughter season, and the ewes were sampled at government-designated slaughter and processing sites, in strict accordance with the “Regulations on the Administration of Livestock and Poultry Slaughter in Qinghai Province”, as well as the relevant regulations on experimental animal welfare and ethical review by the Ethics Committee of the Laboratory Animal Management Committee of Qinghai Academy of Animal Husbandry and Veterinary Sciences (protocol code QHMKY-2020-09 on 24 June 2020).

### 2.2. Library Construction and Sequencing

Total RNAs were isolated from the ovary of Tibetan sheep using the RNAprep pure Tissue Kit (Tiangen, Cat.DP431). RNA-seq libraries were constructed using three micrograms of highly qualified total RNAs (RNA integrity > 8) by TruSeq RNA Library Prep Kit v2 (Illumina, Cat. RS-122-2001) [12]. The RNA-seq libraries were then sequenced on the Illumina Hiseq X Ten platform to generate pair-end 150 bp reads.

### 2.3. Analysis of RNA-Seq

Trim_galore (version 0.6.6, https://github.com/FelixKrueger/TrimGalore (accessed on 15 March 2022)) and cutadapt (version 3.5, https://cutadapt.readthedocs.io/en/stable/ (accessed on 15 March 2022)) were used to remove adapter contamination and 20 nt low-quality reads from the 5′ end of RNA-seq reads. After quality control, three samples (high-altitude stress: H4 and low-altitude stress: H5–6) did not obtain enough reads (<1000,000 reads) and, thus, they were removed. Nine high-quality samples (high-altitude stress: H1–3, 5–6 and low-altitude stress: H1–4) were retained for further analysis. The cleaned RNA-seq reads were then mapped to Ovis aries genome-oviAri4 (https://hgdownload.soe.ucsc.edu/goldenPath/oviAri4/bigZips/ (accessed on 14 March 2022)) by HISAT2 (version 2.2.1) with default parameters [17]. Mapped reads were counted to individual genes by featureCounts (version 2.0.1) using default parameters [18]. Differentially expressed genes were identified by R package-DESeq2 (version 1.32.0) using the criteria: |log_2_Fold Change| > 1 and false discovery rate (FDR, Benjamini-Hochberg methods) < 0.05 [19]. Sheep amino acid sequences were annotated by eggNOG-MAPPER [20]. Gene ontology and KEGG pathway analyses were performed by TBtools [21]. Protein–protein interaction networks were analyzed using STRING (version 11.5) with default parameters [22]. Read coverage was visualized by IGV (version 2.8.9) [23]. Transcription factors were predicted by AnimalTFDB3.0 with default parameters [24].

### 2.4. Analysis of Alternative Splicing

RNA-seq reads were qualified and mapped using the same strategy as in the section: Analysis of RNA-seq. The output sam files of HISAT2 were converted to bam files and sorted by Samtools (version 1.9) [25]. The sorted bam files were then normalized according to RPKM by deepTools (version 3.5.1) [26]. The gtf files were assembled from the normalized bam files using StringTie (version 2.1.6) [27]. The merged gtf files were generated by TACO (version 0.7.3) [28]. The merged gtf was annotated to the sheep genome using gffcompare (version 0.11.2) [29]. Differential alternative splicing events were identified by rMATS (version 4.1.2) using the criteria: false discovery rate (FDR) < 0.05, |IncLevelDifference| > 0.1 [30].

### 2.5. Experimental Validation

RT-PCR and qRT-PCR experiments were conducted as described in previous studies [12,31], and the results of RT-PCR were visualized in 1% agarose gels. The primers used in this study were listed in Appendix A.

## 3. Results

### 3.1. High-Altitude Stress Affected Gene Expression in Tibetan Sheep

To elucidate the effect of high-altitude stress on Tibetan sheep ovary development, we performed transcriptome analysis of Tibetan sheep ovaries at different altitudes. Nine RNA-seq libraries were constructed and sequenced using total RNA isolated from high- and low-altitude Tibetan sheep ovaries. Illumina RNA-seq generated a total of 221 million reads, and over 96.56% of the reads were mapped to the sheep genome (Appendix A). The ovaries of Tibetan sheep at low altitude (3000 m.a.s.l, Haiyan, Qinghai, China) were used as the control group, and the ovaries of Tibetan sheep at high altitude (4400 m.a.s.l, Yushu, Qinghai, China) were used as the high-altitude stress group.

A total of 18,561 genes (>30 reads per gene) were expressed in the ovaries of Tibetan sheep. In total, 104 genes were identified as differentially expressed genes (DEGs) under high-altitude stress (Appendix A). The number of downregulated genes (64) in high-altitude ovaries was 1.60 times higher than that of upregulated genes (40) (Figure 1a, Appendix A). To understand whether the repression of gene expression at high altitude is a global event or occurs only in certain gene groups, we analyzed the expression patterns of all expressed sheep genes at different altitudes. The results showed that the mean expression counts of all sheep genes did not show any clear pattern between high- and low-altitude sheep (Appendix A). Therefore, we believe that high-altitude stress is not a globally regulated expression of Tibetan genes but is more likely to affect the expression of certain gene groups.

### 3.2. High-Altitude Stress Repressed the Expression of Follicle Development Genes

To further understand the function of high-altitude-related DEGs, we performed KEGG and gene ontology (GO) analyses on the DEGs. KEGG analysis revealed that DEGs were mainly enriched in environmental information processing and cancer-related pathways (Appendix A). The results of GO analysis showed that high-altitude stress-related DEGs were over-represented in several biological processes, including regulation of vascular development, ovulation cycle, ossification, response to cAMP, regulation of cytokine production, response to lipopolysaccharide, positive regulation of cell motility, positive regulation of cell migration, epithelial cell proliferation, cellular responses to growth factor stimulation and cell adhesion (Figure 1b). Importantly, these processes are closely related to the regulation of ovarian follicle development or ovulation [32,33,34,35,36,37,38]. A total of 36 DEGs were identified from these GO terms, many of which were well-characterized ovarian follicle development or ovulation-regulating genes (Appendix A). For example, lymphatic vessel endothelial receptor-1 (*LYVE1*) is a molecular marker of lymphatic vessels and is used to distinguish lymphatic vessels from blood vessels [39]. *LYVE1* has been reported to be involved in the development of the ovarian lymphatic vasculature and is critical for follicular fluid control [33,40]. In the present study, *LYVE1* was one of the most significantly repressed genes under high-altitude stress (Appendix A), suggesting that high-altitude stress may inhibit the development of the ovarian lymphatic system and follicular fluid through suppressing the expression of *LYVE1*. Another example is *ADAMTS-1* (a disintegrin and metalloproteinase 1 with a thrombospondin motif); *ADAMTS-1* is a central regulator of structural remodeling during ovarian follicle growth, and knockout of *ADAMTS-1* results in a defective lymphatic network in mouse ovary [32,33,41]. The downregulation of *ADAMTS-1* under high-altitude stress unveiled that high-altitude stress may repress the lymphatic development of ovarian follicle via suppressing the expression of *ADAMTS-1*. Interestingly, we also observed that 28 DEGs from these GO terms were downregulated under high-altitude stress (Figure 1b, Appendix A) and, together with their functional enrichment in ovarian follicle development (Figure 1b), we speculate that high-altitude stress may inhibit ovarian follicle development in Tibetan sheep.

### 3.3. Hormone Signaling Transcription Factors Were Central Regulators in Response to High-Altitude Stress

Transcription factors (TFs) are key regulators in environmental stress responses [42]. To understand the relationship between TFs and high-altitude stress, we characterized all sheep TFs and overlapped them with DEGs. A total of seven TF encoding genes were identified in DEG, which were early growth response 1 (*EGR1*), Fos proto-oncogene, AP-1 transcription factor subunit (*FOS*), FosB proto-oncogene, AP-1 transcription factor subunit (*FOSB*), Krüppel-like factor 9 (*KLF9*), zinc finger and BTB domain containing 16 (*ZBTB16*), MYCN proto-oncogene, BHLH transcription factor (*MYCN*), and nuclear receptor subfamily 4 group A member 3 (*NR4A3*) (Figure 1c, Appendix A). Five of them, including *EGR1*, *FOS*, *FOSB*, *KLF9*, and *ZBTB16*, were downregulated under high-altitude stress (Figure 1c, Appendix A). More importantly, *EGR1*, *FOS*, and *FOSB* have been reported to positively regulate ovarian follicle development [38,43,44].

Transcription factors are components of hormone signal transduction pathways. To further explore the central regulatory network of high-altitude stress-related DEGs, we used STRING to perform protein–protein interaction network (PPI) analysis on DEGs encoding genes [22]. PPI network analysis revealed that the number of edges for the PPI-related DEG was 31, which is much larger than the expected number of edges (12) with a PPI enrichment *p*-value of 3.18 × 10^−6^ (Figure 1d). The significant enrichment of the PPI network further highlights that there were more interactions between proteins encoded from DEGs than expected, and that these proteins may be biologically partially connected as a group [22]. To further explore the biologically connected group of these factors, we performed local network clustering using STRING. The results showed that c-fos/v-fos, and protein fosb, and the basic region leucine zipper of hormone ligand binding were significantly enriched biologically groups, with FDR values equal to 4.46 × 10^−5^ and 2.40 × 10^−4^, respectively (Figure 1e, Appendix A). A total of six genes were identified from these two biologically connected groups, which were *FOS*, *DUSP1*, *FOSB*, *NR4A3*, *FKBP5*, and *EGR1* (Figure 1e, Appendix A). Interestingly, four of them (*FOS*, *DUSP1*, *FOSB*, and *EGR1*) were responses to cAMP and they were downregulated under high-altitude stress (Figure 1f, Appendix A). cAMP acts as a secondary messages of luteinizing hormone (LH) and follicle-stimulating hormone (FSH) in the ovary [34]. cAMP is not only responsible for activating primordial follicles into follicle growth, but also plays an important role in the growth and development of all other ovarian follicular stages [35]. Therefore, we suggest that altitude stress may inhibit follicle development by impairing follicle-stimulating hormone signaling transduction in Tibetan sheep.

### 3.4. Differential Expressed Genes Were Validated by qRT-PCR Experiments

To validate high-altitude stress-associated genes, qRT-PCR analysis was performed to evaluate the differential expression patterns of key DEGs under high-altitude stress. The expression of luteinizing hormone (LH)- and follicle-stimulating hormone (FSH)-related genes: *EGR1*, *FKBP5, DUSP1*, and *FOS* were significantly downregulated under high-altitude stress by qRT-PCR (Student *t*-test *p*-value: *EGR1*, *p* = 1.2 × 10^−2^; *FKBP5*, *p* = 9.4 × 10^−3^; *DUSP1*, *p* = 9.7 × 10^−3^; *FOS*, *p* = 6.1 × 10^−3^; Figure 2a–d). In addition, the expression of lymphatic vasculature and network marker genes: *LYVE1 and ADAMTS-1* were also significantly repressed under high-altitude stress by qRT-PCR (Student *t*-test *p*-value: *LYVE1*, *p* = 6.4 × 10^−3^; *ADAMTS-1*, *p* = 2.8 × 10^−5^; Figure 2e,f). These results suggest that high-altitude stress reduced ovarian follicular development by inhibiting LH and FSH hormone signaling and lymphatic system development.

### 3.5. Alternative Splicing Is Widespread in Tibetan Sheep

Alternative splicing (AS) is an important post-transcriptional regulation mechanism that regulates mRNA fate and diversifies the proteome [12]. Alternative splicing in Tibetan sheep has not been characterized, and whether high-altitude stress affects alternative splicing in livestock remains elusive. To identify alternative splicing in Tibetan sheep and decipher the relationship between AS and high-altitude stress, we performed alternative splicing analysis using RNA-seq by bioinformatics software. Among the 18,561 expressed genes (>30 reads per gene), about 52.01% (9654/18,561) of Tibetan sheep genes underwent alternative splicing events (40,773 AS events). Next, we classified expressed genes into four types based on the number of AS events per gene: none (AS events = 0), low (AS events = 1), moderate (AS events = 2–10), and high (AS events = 2–10) events > 10 (Figure 3a). AS was prevalent in Tibetan sheep, as more than 70% (6938/9654) of AS-containing genes had two or more AS events (Figure 3a). Therefore, these results suggest that alternative splicing is widespread in Tibetan sheep.

### 3.6. High-Altitude Stress Induced Differential Alternative Splicing in an AS-Type-Specific Pattern

Alternative splicing events can be classified into five types: skipped exon (SE), retained intron (RI), alternative 5′ splice site (A5SS), alternative 3′ splice site (A3SS), and mutually exclusive exon (MXE) [30]. In Tibetan sheep, SE was the most dominant AS type (63.8%), and RI was the second largest AS type (10.7%). The percentages of the other three AS types (A5SS, A3SS, and MXE) were much closer to each other, and they each accounted for about 8% of the total number of AS events (Figure 3b, Appendix A). Importantly, 917 AS events corresponding to 671 genes were identified as high-altitude stress-induced differential alternative splicing (DAS) in Tibetan sheep, accounting for 6.94% (670/9654) of AS-containing genes (Figure 3c, Appendix A, Appendix A). In particular, the distribution of AS types from DAS was different from that of total AS (Figure 3b,d). Unlike SE in total AS, the most dominant AS type was RI in DAS, and the percentage of RI in DAS (35.6%) was 3.33 times higher than that in total AS (10.7%) (Figure 3b,d). The percentages of A5SS and A3SS in DAS (A5SS: 16.5%, A3SS: 16.4%) almost doubled compared to total AS (A5SS: 7.9%, A3SS: 8.7%) (Figure 3b,d). In contrast, the percentages of both SE and MXE were significantly decreased in DAS compared with total AS (SE: AS vs. DAS, 63.8% vs. 27.3%; MXE: AS vs. DAS: 8.9% vs. 4.4%) (Figure 3b,d). Overall, high-altitude stress induced RI, A5SS, and A3SS-type AS events but suppressed SE and MXE (Figure 3b,d). Thus, these results suggest that high-altitude stress induces DAS in an AS-type-specific pattern.

To explore whether high-altitude stress-related DAS have global expression patterns, we performed expression analysis of different types of DAS with all expressed genes. The expression of genes containing ES and MXE types of DAS was significantly reduced compared with all expressed genes (Figure 3e). In contrast, there was no global expression pattern change for high-altitude stress associated with RI, A5SS, and A3SS (Figure 3e). Therefore, these results suggest that the expression of ES- and MXE-type DAS genes may be regulated at the transcriptional level, while the expression of RI, A5SS, and A3SS genes may be regulated at the post-transcriptional level. Overall, DAS associated with high-altitude stress may affect gene expression in an AS-type-dependent manner.

### 3.7. C_2_H_2_-Type Zinc Finger Transcription Factors Enriched in High-Altitude Stress-Related DAS

To decipher the function of high-altitude stress-related DAS, KEGG analysis of DAS genes was performed. The results showed that high-altitude stress-related DAS genes mainly acted on RNA splicing, transcription factors, chromosomes and related proteins, DNA repair and recombination, and ribosome biogenesis (Figure 3f, Appendix A).

To understand the role of transcription factors on altitude stress, we analyzed the relationship between TF and DAS. About 11.33% of TFs (76/671) had DAS events associated with high-altitude stress (Figure 4a, Appendix A). Surprisingly, 73.78% of the DAS-related TFs (56/76) were C_2_H_2_-type zinc finger (ZNF) TFs (Figure 4b, Appendix A). Therefore, we speculate that C_2_H_2_-type zinc finger (ZNF) TFs may be regulated by high-altitude-induced DAS.

### 3.8. High-Altitude Stress-Associated Retained Introns Were Enriched in RNA Processing Factors

As retained intron is the major AS type induced by high-altitude stress, we next analyzed whether different types of DAS have distinct enriched biological groups by STRING’s protein–protein interaction network (PPI) analysis. PPI analysis revealed no significant associated biological groups in high-altitude stress-related A3SS and MXE (PPI enrichment *p*-values: 0.101 and 0.58). High-altitude stress-related ES and A5SS had significant PPI-enriched *p*-values (PPI-enriched *p*-values: 0.0138 and 0.00529), but they had no significant biological association groups. However, high-altitude stress-associated RI did have significant PPI enrichment and enrichment of biological groups (Figure 4c). PPI network analysis revealed that the high-altitude stress-related RI had a number of edges of 186, which was much larger than the expected number of edges (112) for a PPI enrichment *p*-value of 9.69 × 10^11^. Local network cluster enrichment of high-altitude stress-related RIs revealed that RNA splicing-related terms, including the thrap3/bclaf1 family and mRNA 3-terminal processing, were biologically significantly enriched, with FDR values of 3.50 × 10^−3^ and 7.40 × 10^−3^, respectively (Figure 4d, Appendix A). A total of nine genes related to RNA processing were identified, namely *SRRM2*, *CLK1*, *CLK4*, *TRA2A*, *RBM39*, *ACIN1*, *PCF11*, *CLP1*, and *DDX39B*, including splicing and polyadenylation factors. *CLK4* is a serine- and arginine-rich (SR) protein-encoding gene that regulates pre-mRNA splicing [45]. According to RT-PCR results, the ratio of RI transcripts to normal transcripts increased from 1.62 at low altitude to 6.0 at high altitude (Figure 4e,f). Thus, high-altitude stress induced the RI of *CLK4*. Overall, these results suggest that high-altitude stress may affect RNA processing by inducing DAS.

## 4. Discussion

High-altitude stress is the main environmental stress for the growth and production of Tibetan sheep. The molecular basis of high-altitude stress response mechanism of Tibetan sheep remains elusive. In the present study, comparative transcriptome analysis was performed using RNA-seq on ovarian tissues of Tibetan sheep exposed to high altitude and low altitude. A total of 104 genes were differentially expressed in high- and low-altitude Tibetan sheep ovaries (Appendix A). Of these, 61.54% (64/104) of DEGs (64/104) were downregulated under high-altitude stress, and the suppression of gene expression under high-altitude stress did not show a global gene expression pattern but may only occur in specific gene groups (Appendix A). Gene ontology analysis revealed that high-altitude stress-associated DEGs were over-represented in biological processes related to ovarian follicle development or ovulation regulation (Figure 1b).

More importantly, network analysis revealed that hormone-responsive c-fos/v-fos family transcription factors were significantly enriched in the high-altitude stress-related gene expression regulatory network (Appendix A, Figure 1e,f). These regulators were *FOS*, *DUSP1*, *FOSB*, and *EGR1* (Appendix A). In mice, EGR1 significantly induced *ADAMTS-1* expression, and an *EGR1 binding site* (*EBS*) was found in the *ADAMTS-1* promoter [46]. Furthermore, EGR1 regulates *ADAMTS-1* expression in a dose-dependent manner [46]. In the present study, the expressions of *EGR1* and *ADAMTS-1* were both significantly downregulated in Tibetan sheep ovaries under high-altitude stress (Appendix A, Figure 2a,f). Taken together, EGR1 and ADAMTS-1 positively regulate follicular development, and EGR1 is an important component of FSH signaling [43,47]. Therefore, high-altitude stress may inhibit follicular development by impairing the FSH–EGR1–ADAMTS-1 signaling pathway. The expression level of *TNFα-induced protein 6* (*TNFAIP6*) is highly increased in follicular cells, and mutation of *TNFAIP6* causes infertility in mice [44,48]. The expression of *TNFAIP6* in granulosa cells is regulated by the cAMP response element [44]. In this study, high-altitude stress suppressed the expression of *TNFAIP6* (Appendix A). As the expression of *TNFAIP6* is induced by LH, and *TNFAIP6* is responsive to cAMP [44], we speculate that high-altitude stress may also inhibit follicular development by impairing the LH–cAMP–TNFAIP6 signaling pathway. Overall, the results of this study suggest that hormonal signaling plays a key role in Tibetan sheep ovarian tissue response to high-altitude stress.

Alternative splicing is one of the important post-transcriptional regulatory mechanisms in response to environmental stress [12]. Abnormal expression of splicing factors and aberrant splicing are closely related to the development of ovarian cancer [49]. In this study, high-altitude stress induced 917 AS events in Tibetan sheep (Figure 3c). In particular, RI but not ES was the predominant AS type in high-altitude stress-related DAS (Figure 3d). Taken together with RI results from aberrant splicing [50], these results indicated that high-altitude stress induced aberrant splicing in Tibetan sheep ovaries. Cdc2-like kinase (CLK4) plays a role in regulating pre-mRNA splicing [45]. In this study, high-altitude stress induced RI and, thus, increased the aberrant splicing isoform of *CLK4* (Figure 4e,f). Taken together with mutation of *CLK4* reducing cell proliferation in ovaries [45], we speculated that high-altitude stress may repress cell proliferation by increasing abnormally spliced isoform of *CLK4*. Overall, our results unveiled that high-altitude stress indeed affects the follicular development at both the transcriptional and post-transcriptional levels.

## 5. Conclusions

Tibetan sheep live in the harsh environment of Qinghai–Tibet Plateau and high-altitude stress significantly reduces their fertility. Understanding the molecular mechanism of Tibetan sheep responding to high-altitude stress will provide important genetic resources for the research and application of high-altitude response in sheep. In this study, we characterized 104 high-altitude stress-related genes, and follicular-development-related genes were significantly downregulated under high-altitude stress. More importantly, LH and FSH signaling-related transcription factors were significantly enriched in the differentially expressed genes induced by high-altitude stress. In addition, 917 AS events were characterized in response to high-altitude stress and an increase in abnormally spliced isoforms. High-altitude stress induces RI but inhibits ES-type AS events. Taken together, our findings provide evidence that high-altitude stress represses follicle development genes at both the transcriptional and post-transcriptional levels.

## Figures and Tables

**Figure 1 animals-12-02812-f001:**
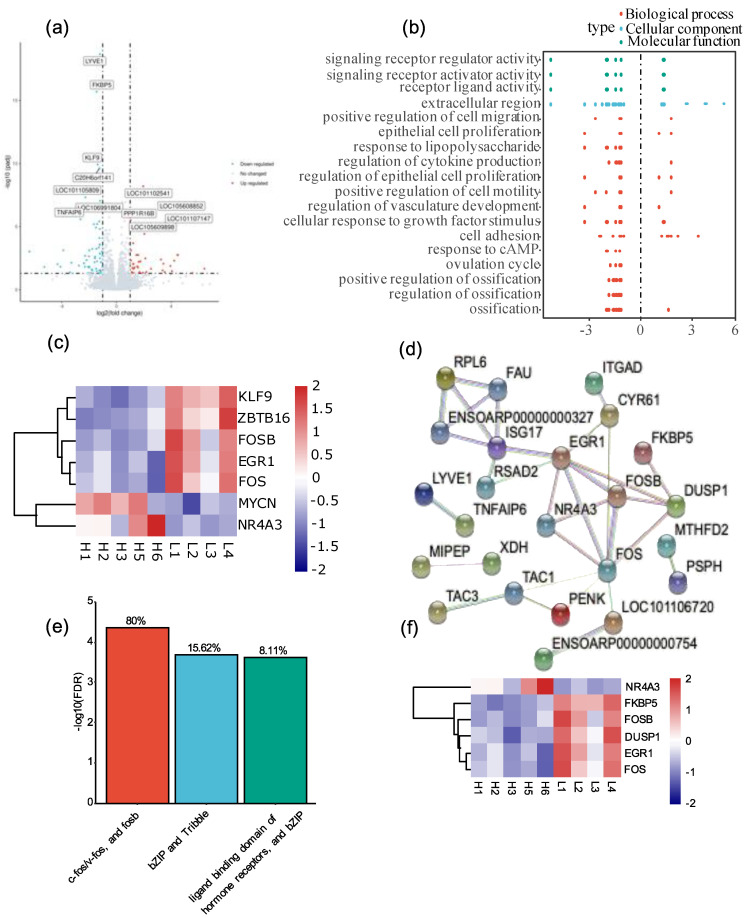
High-altitude stress induced differential expression of follicle development genes. (**a**) The volcano plot highlights the most significant differential expressed genes of ovarian tissues under high-altitude stress in Tibetan sheep; the significant difference cut-off criterion is *p* < 0.05 and |log2Fold Change| > 1; blue dots, the significantly downregulated genes; red dots, upregulated genes under high-altitude stress. (**b**) Gene ontology enrichment of differential expressed genes; the dots of different color represent the classification group of the function. (**c**) The heatmap of high-altitude stress-associated transcription factors. (**d**) The protein–protein-interaction (PPI) network of DEGs; each of the nodes represents a protein. (**e**) The functional enrichment clusters in the DEGs’ PPI network, the horizontal axis is the description of local network cluster. The left vertical axis is the FDR. The percentage on the column is the ratio of the number of genes clustered to each entry to the total number of genes in the entry. (**f**) The heatmap of high-altitude stress-associated transcription factor from PPI network.

**Figure 2 animals-12-02812-f002:**
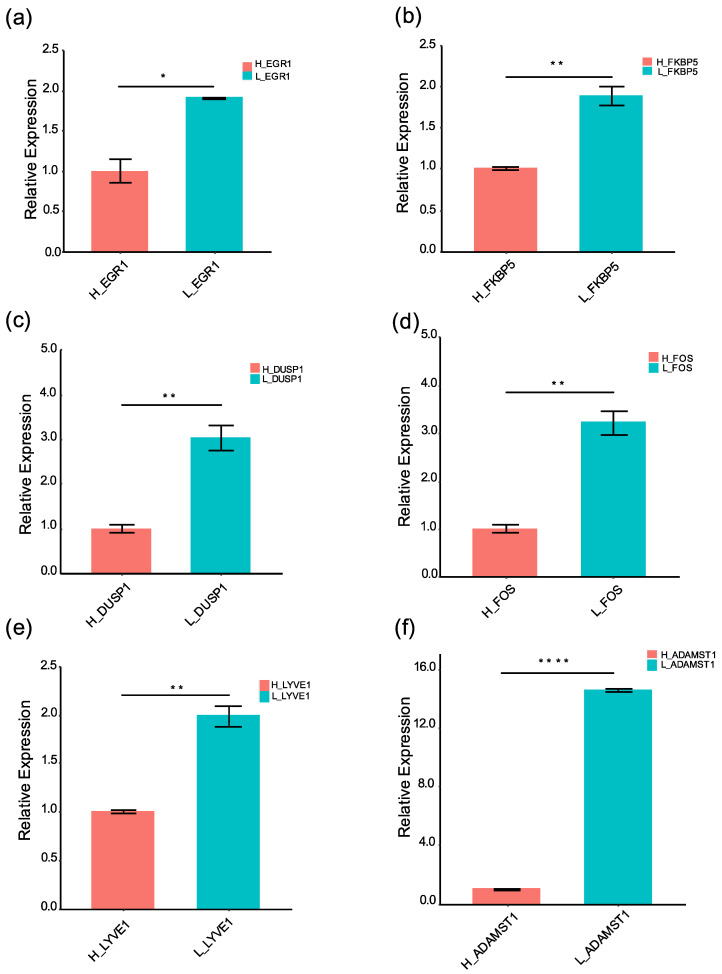
High-altitude stress reduced expression of hormone signaling transcription factors and lymphatic development marker genes. The relative expression pattern of EGR1 (**a**), FKBP5 (**b**), DUSP1 (**c**), FOS (**d**), LYVE1 (**e**), and ADAMST1 (**f**) in Tibetan sheep ovaries at low altitude and high altitude. The relative expression of each gene was detected by quantitative reverse transcription-PCR and normalized to GAPDH expression using the 2^–ΔΔCt^ method. Error bars represent standard deviation of three biological replicates. *Y*-axis: relative expression levels; *X*-axis: genes; H: high altitude; L: low altitude; “*” denoted the Student *t*-test *p* < 0.05; “**” denoted the Student *t*-test *p* < 0.01; “****” denoted the Student *t*-test *p* < 0.0001.

**Figure 3 animals-12-02812-f003:**
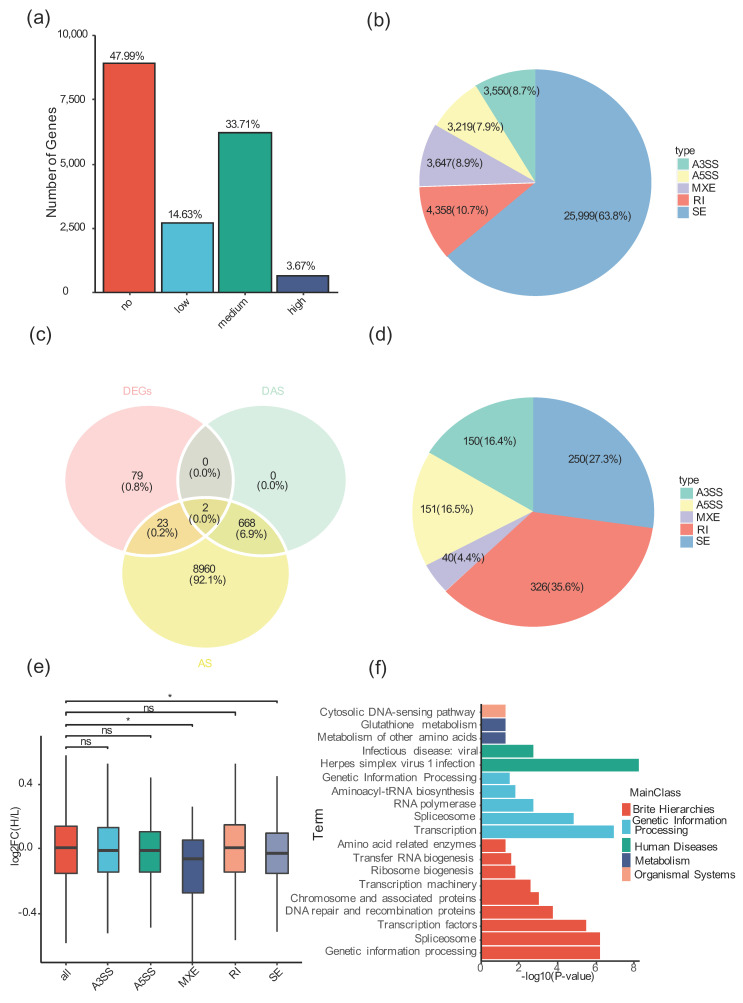
High-altitude stress induced the retained intron type of alternative splicing events in Tibetan sheep ovary. (**a**) Bar plot of all expressed genes classified into four groups: gene expressed but lacking AS events is defined as no, the low-frequency AS gene group was only 1 AS event, the medium-frequency AS gene group was 2 to 10 AS events, the high-frequency AS gene group was ≥11 AS events. (**b**) Pie chart of the five different types of AS and their frequency in the RNA-seq data sets of Tibetan sheep grown in different altitudes areas. A3SS, alternative 3′ splice site; A5SS, alternative 5′ splice site; MXE, mutually exclusive exons. (**c**) The Venn diagrams indicate overlaps among DEGs, DAS genes, and AS genes under high or low altitude condition. (**d**) Pie chart of the five different types of DAS. (**e**) Expression box plot of five different types of DAS genes. The Wilcoxon signed-rank test was applied to evaluate the statistical significance of difference between two samples, and *p* values are indicated above (* denotes *p* ≤ 0.05). ns: represents no significance. (**f**) KEGG pathway analysis of RI type DAS genes. The left vertical axis is the specific name of the KEGG pathway, and the horizontal axis is the significance. The bar of different color represents the classification name of the pathway.

**Figure 4 animals-12-02812-f004:**
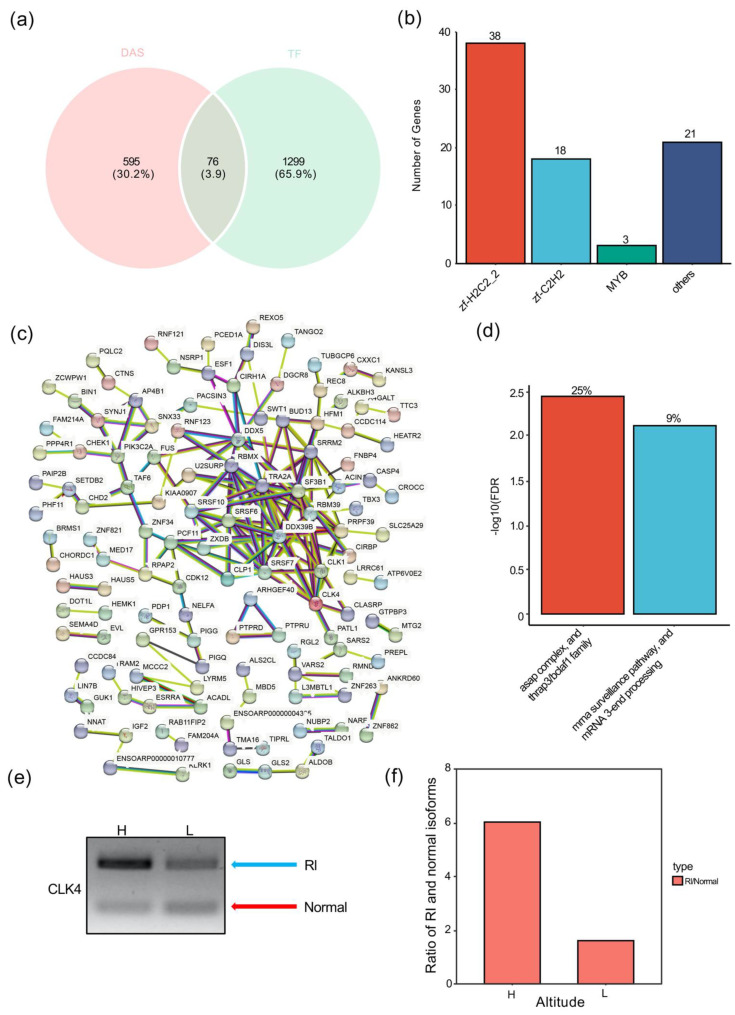
High-altitude stress-induced DAS genes mainly enriched in C_2_H_2_ transcription factors and splicing factors. (**a**) The Venn diagrams indicate overlaps among DAS genes and TF genes. (**b**) The distribution of alternatively spliced TF genes. (**c**) The PPI network of DASs. (**d**) The functional enrichment clusters in the retained intron type of PPI network. (**e**) Expression of retained intron (RI) transcripts and normal transcripts (Normal) by RT-PCR. (**f**) The ratio between RI transcripts and normal transcripts with the gray value calculated with Gel-Pro Analyzer 4.

## Data Availability

The datasets from Illumina have been deposited on the NCBI website under the SRA Bioproject accession number: PRJNA871383.

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
