# Peer review of "High-Altitude Stress Orchestrates mRNA Expression and Alternative Splicing of Ovarian Follicle Development Genes in Tibetan Sheep"

_animals, 2022, doi:10.3390/ani12202812_

Round 1

Reviewer 1 Report

High-altitude stress is one of major environmental stresses to reduce fertility of livestock in Qinghai-Tibetan plateaus and how the high-altitude stress affected the development of ovarian follicles are largely unexplored. In this manuscript, Li et al used comparative transcriptomics to systematically analyzed how high-altitude stress regulated the expression of ovarian follicular genes. The identification of high-altitude stress inhibited the expression of key ovarian follicular developmental genes including marker genes: ADAMTS-1 and LYVE as well as LH/FSH related transcription factors could provide certain explanation for high-altitude stress associated infertility. In addition, the authors also identified alternative splicing events and dynamics of AS events under high-altitude stress. Overall, the results could provide certain novelty and contribution for understanding how the high-altitude stress involves in ovarian follicular development associated gene expression regulation.

Generally, the manuscript is well written and structured. The scope and the topic are suitable for Animals. However, there are several points, which require clarification.

  1. Line60: “The annual output value of Tibetan sheep industry in Qinghai Province alone reaches 12.70 billion RMB or ~1.95 billion US dollars.”

The unit of RMB should be the international standard CNY.

  1. Line64: “The Tibetan ewes only give birth to one time a year, and the mortality rate of neonatal lamb is ~ 6%.”

The language of canonical sentences should be used to make it easier for readers to understand.

  1. Line87: “High altitude stress induced retained intron but inhibiting skipped exon events.”

Please provide relevant evidence to testify the content of the testimony.

  1. Line98-102: “Tibetan sheep ovary tissues under high-altitude stress were collected in July 2020 at the Zangbala Breeding Farm (average altitude 4400 m.a.s.l) in Yushu City (E 97â—¦02′; N 33â—¦00′), Qinghai Province, China. Likewise, low-altitude Tibetan sheep ovary tissues were col-lected in July 2020 from the Tibetan sheep breeding cooperative in Jintan Township, Haiyan County (E 100â—¦98′; N 36â—¦90′), Qinghai Province, China.”

The author should added the altitude status of low-altitude group.

  1. Line102-103: “To ensure high similarity in the growth and development stages of Tibetan sheep, all samples were collected from two-year-old nulliparous ewes.”

The transcriptome is not always the same at each stage of growth in sheep, so why was it sampled at two-year-old? Is it a time for sampling suitable for the purpose of the study?

  1. Line117-118: “After quality control, three low quality samples were removed and nine high-quality samples were retained for further analysis.”

Which altitude group were the samples removed from? What is the reason for removing these samples?

  1. Line130: “The sorted bam files were then normalized by deep Tools (version 3.5.1).”

Which way exactly? Is it RPKM, FPKM or TPM? Please give an explanation.

Reviewer 2 Report

its a very interesting subject the results added and explained many question , i strongly agree to publish it, but few comments below may be useful:

-          In the introduction the authors should add their objectives rather than their findings please rewrite the last sentence

-          Samples or tissues collection procedure is not clear please add the collection procedure 

Reviewer 3 Report

The manuscript “High-altitude stress orchestrates mRNA expression and alternative splicing of ovarian follicle development genes in Tibetan sheep” analyzed the transcriptome dynamics of Tibetan sheep ovaries under high-altitude stress. The manuscript is well written, with the need for some important details in methodology.

In the abstract:

1. I suggest that the authors rewrite the conclusions, indicating more specifically what the results conclude.

In the material and methods:

1. I suggest the authors detail how the animals were selected. In addition, the authors must explain whether they were animals resulting from a scheduled slaughter or not.

2. I suggest entering information about ethics committee and animal handling.

3. How were the ovaries processed? How were they transported? Was there any pre-treatment on them before they were placed in the liquid nitrogen cylinder? It is interesting to detail all this information.

4. Was there no statistical analysis to compare the data?
